# Protective Effects of Epigallocatechin Gallate for Male Sexual Dysfunction in Streptozotocin-Induced Diabetic Rats

**DOI:** 10.3390/ijms23179759

**Published:** 2022-08-28

**Authors:** Andy C. Huang, Ta-Chuan Yeh, Nien-Chin Wu, Chien-Yu Yeh, Pei-Hua Lin, Kuei-Ying Yeh

**Affiliations:** 1Institute of Traditional Medicine, School of Medicine, National Yang Ming Chiao Tung University, Taipei 112304, Taiwan; 2Department of Urology, Taipei City Hospital Ren-Ai Branch, Taipei 103212, Taiwan; 3Department of Psychiatry, Tri-Service General Hospital, National Defense Medical Center, Taipei 114201, Taiwan; 4Department of Physical Therapy, HungKuang University, No. 1018, Sec. 6, Taiwan Boulevard, Shalu District, Taichung 43302, Taiwan; 5School of Nursing, National Defense Medical Center, Taipei 114201, Taiwan

**Keywords:** diabetes mellitus, EGCG, LH, PDE5a, sexual dysfunction, streptozotocin

## Abstract

Sexual dysfunction is a common problem for men with diabetes. Epigallocatechin gallate (EGCG) is known to ameliorate erectile function in aging rats. However, there has not yet been a report to evaluate its effects on diabetic male rat sexual behavior in the literature. In this study, we investigated the effects of EGCG on male sexual behavior in diabetic rats. Diabetic rats were induced by a single intraperitoneal injection of 65 mg/kg of streptozotocin. After streptozotocin injection for one week, animals were then orally treated with 40 mg/kg of EGCG or vehicle. Copulatory behavior and fasting blood glucose levels were recorded before treatment, as well as 7 and 14 days after treatment. Serum LH, testosterone, and PDE5a levels were measured by EIA assay after the last behavioral test. Data showed that diabetic rats who had diminished sexual functions demonstrated significantly increased latencies in mount, intromission, and ejaculation, as well as significant decreases in frequencies of intromission and ejaculation, compared to non-diabetic controls, indicating sexual function recovery. Lower blood glucose levels were also found in diabetic rats after EGCG treatment. Additionally, the lower LH and higher PDE5a levels in diabetic rats than controls were also noted. The findings declared that EGCG had a protective effect on male sexual behavior in diabetic rats.

## 1. Introduction

Diabetes mellitus (DM) is a chronic metabolic disease characterized by elevated levels of blood glucose. The International Diabetes Federation estimates that approximately 463 million adults (age 20–79 years) were living with diabetes in 2019, and this number is expected to rise to 700 million by the year 2045, according to the International Diabetes Federation [1]. DM can lead to many complications, including retinopathy, nephropathy, and macrovascular disease [2]. Moreover, sexual dysfunction is also a common problem in men who have diabetes [3,4]. A clinical study reported that men with diabetes might experience all aspects of sexual dysfunction, including loss of sexual libido, ejaculatory dysfunction, and erectile problems [4]. Animal studies have demonstrated that *diabetic* rats present substantial *impairment* in mating behavior and erectile function [5,6,7,8].

Streptozotocin (STZ) is the toxic glucose analog most often used to induce diabetes in rats [9]. Previous studies demonstrate that STZ-induced diabetic male rats had a deficiency in sexual behavior [10,11], and a decreased intracavernous pressure (ICP) in STZ-diabetic rats can also be found, suggesting STZ-induced diabetes rats could be a good DM erectile dysfunction (ED) models [12].

Hormonal levels such as luteinizing hormone (LH), and testosterone (T) play key roles in the regulation of male sexual behavior [13,14,15]. T is a primary male sex hormone and is essential for the facilitation of copulatory behavior in males. Studies have demonstrated that its deficiency after castration eliminates penile erection, and abolishes sexual behavior, and these effects could be reversed by T replacement [16,17].

Phosphodiesterase 5 (PDE5) inhibitors are widely used as a first-line treatment for sexual dysfunction [18]. Under PDE5 inhibitors’ treatment, the latencies of mount, intromission, and ejaculation were significantly increased while the numbers of mount and intromission decreased in diabetic rats [7]. Sildenafil known as the first PDE5 inhibitor has been reported effective in the treatment of sexual dysfunction in diabetic rats [7].

Epigallocatechin-3-gallate (EGCG) is the most abundant and biologically active catechin in green tea [19]. EGCG is well-known for its strong antioxidant effect and has a significant effect on increasing cavernous antioxidants in STZ-induced diabetic-aged rats [20]. It has also been reported that EGCG treatment can improve aging rats’ sexual function via increased penile ICP (intracavernosal pressure), indicating that EGCG may be good for erectile function [21].

Penile erection is a dynamic process, requiring dilation of feeder arterioles and cavernous sinusoids, which depends on the relaxation of the cavernous smooth muscle. ED is closely related to the loss of cavernosal smooth muscle cells [22]. A study has shown that EGCG significantly attenuated the loss of cavernosal smooth muscle in diabetic rats after ten weeks of treatment [23]. These studies [21,23,24] suggest that EGCG has a protective effect on ED in diabetic rats.

As mentioned above, male sexual function could be assessed by many aspects, such as sexual behavior parameters, ICP, PDE5, and hormonal changes. The role of EGCG in animal sexual behavior (e.g., in mounting, intromission, and ejaculation parameters) with diabetes has not yet been investigated. Hence, we designed a novel study to evaluate the effect of EGCG on male sexual behavior in STZ-induced diabetic rats.

## 2. Results

### 2.1. Body Weight Changes

The comparison of changes in body weight revealed a significant effect of group (*F*_2,22_ = 8.80, *p* < 0.01), duration (*F*_2,44_ = 6.40, *p* < 0.01), and interaction between group x duration interaction (*F*_4,44_ = 15.37, *p* < 0.001) (Figure 1). The mean body weights significantly decreased due to STZ injection (Control group vs. DM group, *p* < 0.05), and body weight losses were observed in both DM and DM+EGCG groups. Additionally, mean body weight dramatically decreased in the DM+EGCG group after 14 days of EGCG treatment compared with the DM group at the same time point (*p* < 0.05) (Figure 1).

### 2.2. Blood Glucose Levels

As shown in Figure 2, the ANOVA revealed a significant effect of group (*F*_2,22_ = 75.54, *p* < 0.001), duration (*F*_3,66_ = 82.18, *p* < 0.001), and interaction between group x duration (*F*_6,66_ = 22.53, *p* < 0.001) on the glucose levels in diabetic rats. The post-hoc comparisons revealed that the blood glucose levels significantly increased after STZ injection (Control group vs. DM group, *p* < 0.001), and significantly reduced by 14 days of EGCG treatment (DM group vs DM + EGCG Group, *p* < 0.001). The glucose levels in all groups were not significantly different before the STZ injection.

### 2.3. Copulatory Behavior

As shown in Figure 3, no significant impacts on sexual behavior parameters were found among groups before STZ injection.

For latency analysis, ANOVA revealed a significant main effect of group in ML (latency from the introduction of the female to the first mount; (*F*_2,22_ = 5.55, *p* < 0.05)), IL (latency from the introduction of the female to the first intromission; (*F*_2,22_ = 4.30, *p* < 0.05)), and EL (ejaculation latency, latency from the first intromission to ejaculation; (*F*_2,22_ = 3.92, *p* < 0.05)). ANOVA also showed a significant time effect (*F*_2,44_ = 12.29, *p* < 0.001) in EL. Furthermore, a significant effect of the group x time interaction in ML (*F*_4,44_ = 3.33, *p* < 0.05) and EL (*F*_4,44_ = 2.87, *p* < 0.05) were also detected. The ML, IL, and EL in the DM group were significantly longer than those in the control group (*p* < 0.05). Treatment of the diabetic rats with EGCG for 14 days ameliorated the increased ML, IL, and EL (*p* < 0.01 for all). Moreover, both DM and DM+EGCG groups had longer latencies to mount, intromission, and ejaculation than before STZ injection (*p* < 0.05) (Figure 3A–C).

Copulatory frequency parameters analysis showed a significant effect in IF (*F*_2,22_ = 5.74, *p* < 0.01). In EF, ANOVA revealed differences (*F*_2,22_ = 8.40, *p* < 0.01), and there were interaction (*F*_4,44_ = 5.13, *p* < 0.01) effects. The IF (Figure 3E) and EF (Figure 3F) in STZ-induced diabetic animals were significantly lower than those in controls (*p* < 0.05) but were restored after 14 days of EGCG treatment (*p* < 0.05 and *p* < 0.01, respectively). Analysis between before and after STZ injection revealed that the control group and DM+EGCG animals had a significant increase in EF (*p* < 0.05), while a decreased EF was seen in the DM group (Figure 3F). There were no significant differences in MF (Figure 3D).

### 2.4. Correlation Analysis of Sexual Behavior Parameters and Blood Glucose Levels on Day 14

When plotting the sexual behavior parameters (ML, IL, EL, MF, IF, and EF) against the fasting blood glucose levels for three groups (the vehicle-treated control group, vehicle-treated diabetic group, and EGCG-treated diabetic group), Figure 4F showed there was a significant correlation only between EF and blood glucose levels (r = 0.31, *p* < 0.05) (Figure 4).

### 2.5. Weights of Reproductive Organs after 14 Days of STZ Treatment

One-way ANOVA analysis revealed a significant group difference in seminal vesicle weights (F_2,22_ = 6.50, *p* < 0.01) 14 days after STZ injection; they were significantly lower in both DM (0.2 ± 0.02 g/100.g BW) and DM+EGCG (0.2 ± 0.01 g/100.g BW) groups than those in controls (0.26 ± 0.01 g/100.g BW, *p* < 0.01). No significant difference was found in testicular weight among the groups (Figure 5A). Photographs of representative seminal vesicles of control and diabetic rat are shown (Figure 5B).

### 2.6. Serum Levels of LH, T, and PDE5a

We also investigated reproductive-related hormonal changes, such as LH, T, and PDE5a after EGCG treatment in diabetic rats. As shown in Table 1, the levels of LH in diabetic rats were remarkably lower than those in the control and EGCG-treated diabetic rats (F_2,16_ = 3.84, *p* < 0.05) via post hoc analysis. Interestingly, a significant difference in PDE5a levels was also noted (F_2,16_=5.68, *p* < 0.05). The levels of PDE5a in diabetic rats were higher than those in the control and EGCG-treated diabetic rats, (*p* < 0.01, and *p* < 0.05, respectively). However, there was no significant difference in T levels between all groups (F_2,16_ =1.27, *p* = 0.31).

D/W, distilled water; EGCG, epigallocatechin gallate; LH, luteinizing hormone; PDE5a, phosphodiesterase 5a; STZ, streptozotocin.

## 3. Discussion

In the present study, EGCG ameliorates sexual behavior in STZ-induced diabetic rats. Results showed that EGCG significantly improved male sexual performance in diabetic rats, as revealed by an increase in both IF and EF, and a decrease in EL. Apart from sexual performance, EGCG exerted a beneficial effect on sexual motivation in diabetic rats, which was evidenced by the ML and IL.

Hyperglycemia is the primary pathogenic factor in the development of diabetic complications [25]. Sexual performance in male rats diminished due to high blood sugar induced by STZ injection [5,11]. Diabetic rats had longer mount, intromission, and ejaculation latencies, and fewer mount and intromission frequencies than the control group [10]. Our results showed that DM rats significantly decreased sexual behavior compared to the controls, indicating that hyperglycemia causes a decline in male sexual behavior. In the present study, it was revealed that EGCG has ameliorative effects on blood glucose levels and sexual behavior deficits in STZ-induced diabetic rats. There was a significant relationship between EF and blood glucose levels, indicating that EGCG could increase EF in diabetic rats, possibly associated with decreased blood glucose levels. In our experiment, significantly decreased blood glucose levels were noted in the treatment of diabetic rats with EGCG for 14 days compared to rats in the DM group. However, the blood glucose levels could not revert to normal in DM rats post-EGCG treatment in our study. In summary, STZ injection might cause irreversible damage to the rats’ insulin sensitivity and EGCG could restore some STZ effects.

The reproductive organs’ weight (including the testes, epididymis, and seminal vesicles) of STZ-induced diabetic rats could be observed decreased in literature [10,11]. Our results showed that a decrease in reproductive organs weight was only seen in seminal vesicles in the DM group. The findings are not totally inconsistent with other studies on rats [10,11]. These differences could be explained by the fact that our rats got DM just for short time than other studies [10,11,26]. Lager seminal vesical may be related to higher sexual activity was reported in the human study [27]. A similar finding was also reported in a study by Birkhauser et al. [28], showing that mice with larger seminal vesicles had a remarkably higher intromission rate than mice with smaller seminal vesicles. Our results, consistent with previous studies, also showed that male sexual activity was associated with the volume of seminal vesicles [27,28].

Body weights were dramatically reduced in both groups of STZ-treated rats, with or without EGCG. Meanwhile, it is interesting that diabetic rats treated with EGCG for 14 days had significantly lower mean body weight than non-treated diabetic rats. Our result supports a previous study showing that EGCG has an anti-obesity effect [24,29].

It is well-known that the hormones belonging to the hypothalamic–pituitary–gonadal axis affect the male reproductive system and sexual behavior. LH which controls testosterone secretion from testes, is released by the pituitary gland and is controlled by pulses of gonadotropin-releasing hormone. If bloodstream testosterone levels are low, the pituitary gland will release LH. As the levels of testosterone increase, it will inhibit GnRH and LH release via a negative feedback loop [30]. Testosterone released from testes plays an important role in male sexual behavior. Supplement of T to castrated rats may restore serum T levels, leading to restoration of sexual function [31]. Serum T levels were found declined in male rats who had poor sexual activity compared to controls [32]. Our results showed that EGCG significantly enhanced male serum LH levels, without T levels change in diabetic rats. These results are similar to the findings of a previous study in which Lycium barbarum polysaccharides treatment significantly increased serum LH levels but had no effect on the T levels of male rats exposed to heat [32]. Taken together, in the present research, these findings demonstrated that EGCG treatment may adjust sexual hormones’ secretion and then improve male sexual behavior in diabetes.

Diabetes was associated with an imbalance in oxidative stress leading to erectile dysfunction. Sildenafil which is the most common PDE5 inhibitor is widely used for the treatment of erectile dysfunction [33]. In penile erection process, if cyclic guanosine monophosphate (cGMP) is metabolized by PDE5 enzymes, downstream erectile effects will not exert [33]. PDE5 inhibitors act principally on the nitric oxide (NO)–cGMP signaling pathway and are clinically important in the treatment of erectile dysfunction. A previous study demonstrated that EGCG significantly increased cavernous cGMP, and endothelial NO synthase levels in aged diabetic rats [20]. Moreover, it has been reported that via EGCG treatment, decreased ICP and poor erectile function could be ameliorated in aging rats [21]. In the present study, the serum PDE5 levels were significantly lower in EGCG-treated diabetic rats than those in vehicle-treated diabetes. Furthermore, our behavioral data showed that diabetic rats treated with EGCG had more intromissions. The increase in the number of intromissions has been reported to be an indicator of effective erection [34]. We concluded that EGCG could improve penile erection in diabetic rats. Thus, EGCG may have a similar effect as PDE5 inhibitors, leading to an increased NO–cGMP cyclic activity, which in turn improved intromission behavior that reflected on erectile function in diabetic rats.

## 4. Materials and Methods

### 4.1. Animals

#### 4.1.1. Male

Long-Evans male (8 weeks old) and female (7 weeks old) rats were purchased from the Animal Center of National Science Council, Taipei, Taiwan. Animals were kept in groups of three in a cage (47 × 26 × 21 cm)) in a temperature (22 ± 1 °C)- and humidity (55 ± 10%)-controlled room on an inverted 12 h light:dark cycle (lights off at 11:00 AM). Food and water were available ad libitum. The experimental protocols were approved by the Institutional Animal Care and Use Committee, Hung-Kuang University (Taichung, Taiwan, HK-10614), and all experimental procedures conformed to the National Institutes of Health Guide for the Care and Use of Laboratory Animals. The schematic design of this study is illustrated in Figure 6.

#### 4.1.2. Female

Female Long-Evans rats aged 8 weeks under Zoletil (Virbac, Carrous, France) anesthesia at a dose of 20 mg/kg were ovariectomized and immediately implanted with a 5 mm (internal diameter [ID] = 1.98 mm, and outer diameter [OD] = 3.18 mm) silastic capsule filled with 17 β-estradiol (Sigma-Aldrich, St. Louis, MO, USA). After surgery, females recovered for approximately one week before behavioral testing. To induce sexual receptively on the day of testing, each ovariectomized female was subcutaneously injected with 500 μg progesterone (Sigma), 4 h before testing. Only female rats displaying good sexual receptivity (proceptive behaviors and lordosis) were used.

### 4.2. Drugs

STZ (Sigma) was freshly dissolved in 0.1 M sodium citrate buffer (pH 4.5) at a dose of 65 mg/kg body weight. The STZ dose was chosen based on the previous studies [6,35]. EGCG (Sigma) was dissolved in sterile distilled water and administered orally at a dose of 40 mg/kg/day of EGCG (Sigma) one week after STZ injection. The dose of EGCG was chosen based on a previous study [36].

### 4.3. Behavioral Studies

#### 4.3.1. Copulation Screening of Test Males

To ensure that all males were sexually experienced before the behavioral experiment, copulation screening was carried out during the dark phase of the light/dark cycle (2 h after lights off) under a dim-red light. Each male rat was placed in a circular Plexiglas testing chamber (45 cm diameter), and a sexually receptive female was introduced 3 min later. The screening test was performed over a period of 15 min. The male rats were tested thrice at intervals of 4–5 days, and those that had not ejaculated twice in three testing sessions were not used in subsequent sexual behavioral studies.

#### 4.3.2. Copulation Behavior Testing

Behavioral testing of copulation in male rats was performed between 2:00 PM and 5:00 PM under a dim-red light. For acclimatization to the test environment, each male rat was placed in a circular Plexiglas chamber 3 min before the introduction of a sexually receptive female and was then allowed to copulate for 30 min. The behavioral parameters recorded during the test period were MF (number of mounts), IF (number of intromissions), EF (number of ejaculations), ML (latency from the introduction of the female to the first mount), IL (latency from the introduction of the female to the first intromission), and EL (ejaculation latency, latency from the first intromission to ejaculation). The copulatory behaviors were recorded by an observer.

### 4.4. Induction of Diabetes and EGCG Treatment

One week after screening, the rats were tested for sexual behavior before STZ injection. Then, the rats were randomly divided into three groups: vehicle-treated control group (Control, n = 8), vehicle-treated diabetic group (DM, n = 8), and EGCG-treated diabetic group (DM + EGCG, n = 9).

Diabetes was induced by a single intraperitoneal injection of freshly prepared STZ (65 mg/kg; Sigma). Control animals were intraperitoneally injected with an equivalent volume of sodium citrate buffer. Fasting blood glucose levels were measured from the tail vein before and 72 h after STZ injection, using a glucometer (Contour Plus, Bayer, Germany). Rats exhibiting fasting glucose levels greater than 300 mg/dL after STZ injection were considered as diabetic rats [35,37], and were used in this study.

All the subsequent treatments were done by gavage every day between 9:00 AM and 11:00 AM for 14 consecutive days. Animals in the EGCG-treated diabetic group were administrated at a dose of 40 mg/kg/day of EGCG (Sigma), starting one week after STZ injection. The groups of vehicle-treated control and vehicle-treated diabetes received equal volumes of distilled water. Fasting blood glucose levels were measured again on days 7 and day 14 of EGCG treatment. Body weight and sexual behavioral parameters were recorded before and on day 7 and day 14 of EGCG treatment.

### 4.5. LH, Testosterone, and PDE5a Levels in Serum

After the last behavioral test, animals were *anesthetized* with CO_2_, whereby blood samples and the reproductive organs were collected. For measurement of LH and T concentrations, inferior vena cava blood was harvested into 15-mL test tubes, kept at room temperature for 30 min, centrifuged at 900× *g* for 30 min at 4 °C, and then serum was collected. The levels of LH (Elabscience Biotechnology Inc., Houston, TX, USA), T (Aviva Systems Biology, San Dirgo, CA, USA), and PDE5a (Aviva Systems Biology) were measured by ELISA assays, according to the manufacturer’s instructions.

### 4.6. Reproductive Organ Weight

Bilateral testes and seminal vesicles were collected and weighted. The organ weights were expressed as a percentage of body weight (genital/body weight × 100).

### 4.7. Statistical Analysis

Quantitative data were analyzed for statistical significance using the Statistica software, version 12.0 (Statistica StatSoft, Tulsa, OK, USA). Mean body weight, fasting blood glucose levels, and sexual behavior were analyzed by two-way (Group × Time) ANOVA with repeated measures (time as the repeated factor). The levels of LH, T, and PDE5a in serum, the body weight change, and the weights of reproductive organs were analyzed by one-way ANOVA. Post hoc comparisons used the Fisher LSD test to establish the significance between mean values. Correlations between the sexual behavioral parameters and blood glucose levels were evaluated; *p*-values less than 0.05 were considered statistically significant. All quantitative data are reported as the mean ± SEM.

## 5. Conclusions

In conclusion, this is the first study to demonstrate that EGCG has a beneficial effect on sexual behavior in diabetic male rats. Our findings suggest that EGCG ameliorates male sexual function in diabetic rats probably through elevating LH levels and decreasing PDE5 activity. EGCG has a good safety profile and is already widely used in humans. Our findings provide a possible treatment option for sexual dysfunction in men with diabetes, but further studies are needed to understand the molecular mechanisms involved.

## Figures and Tables

**Figure 1 ijms-23-09759-f001:**
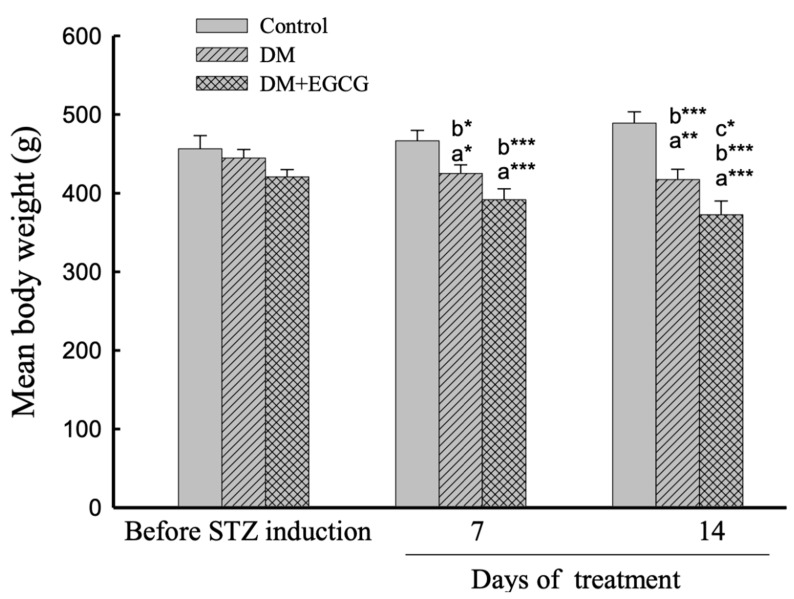
Body weight changes. The data were expressed as mean ± SEM. *, *p* < 0.05; **, *p* < 0.01; ***, *p* < 0.001. (a), compared to the control group at the same time point (7 days or 14 days); (b), compared to the same group before STZ injection; (c), compared to the DM group at the same time point. DM, diabetes mellitus; *EGCG*, epigallocatechin gallate; STZ, streptozotocin.

**Figure 2 ijms-23-09759-f002:**
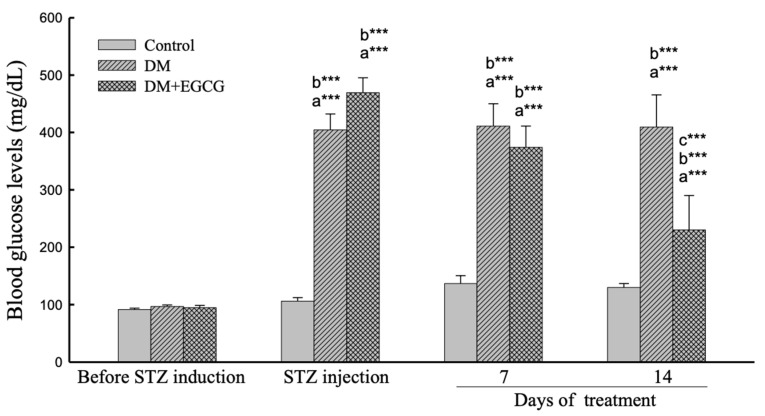
Blood glucose levels. The data were expressed as mean ± SEM. ***, *p* < 0.001. (a), compared to the control group at the same time point (7 days or 14 days); (b), compared to the same group before STZ injection; (c), compared to the DM group at the same time point. DM, diabetes mellitus; *EGCG*, epigallocatechin gallate; STZ, streptozotocin.

**Figure 3 ijms-23-09759-f003:**
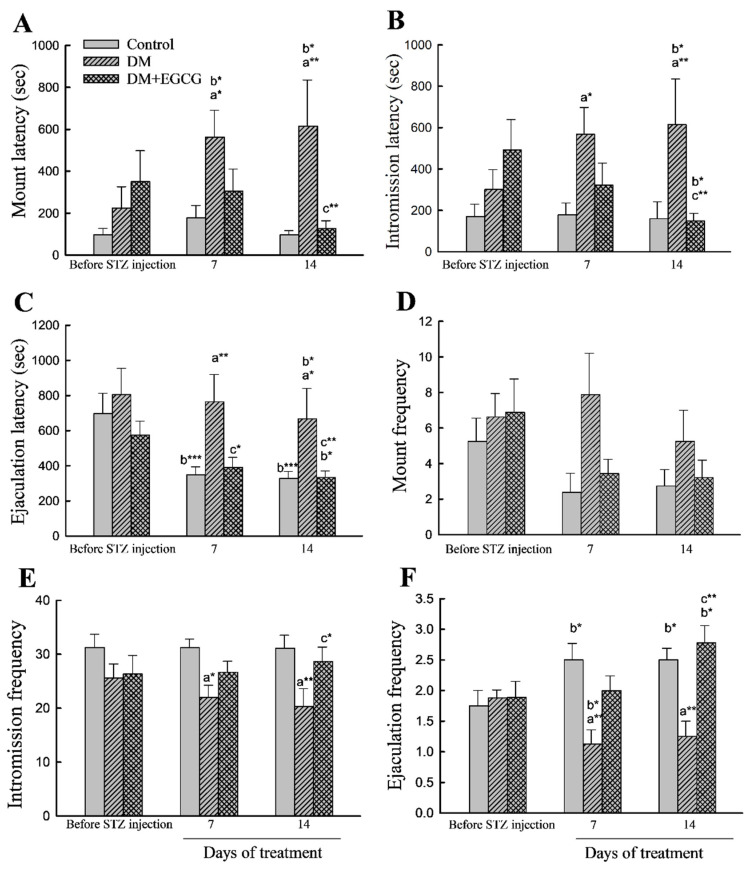
Sexual behavior parameters. (**A**) mount latency; (**B**) intromission latency; (**C**) ejaculation latency; (**D**) mount frequency; (**E**) intromission frequency; (**F**) ejaculation frequency. The data were expressed as mean ± SEM. *, *p* < 0.05; **, *p* < 0.01; ***, *p* < 0.001. (a), compared to the control group at the same time point (7 days or 14 days); (b), compared to the same group before STZ injection; (c), compared to the DM group at the same time point. DM, diabetes mellitus; *EGCG*, epigallocatechin gallate.

**Figure 4 ijms-23-09759-f004:**
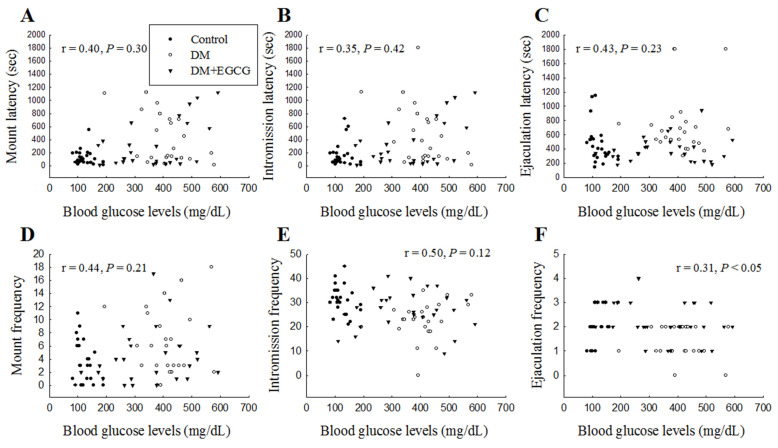
Linear correlation between sexual behavioral parameters (**A**) mount latency, (**B**) intromission latency, (**C**) ejaculation latency, (**D**) mount frequency, (**E**) intromission frequency, and (**F**) ejaculation frequency with blood glucose levels in each group over 14 days of EGCG or distilled water treatment. DM, diabetes mellitus; *EGCG*, epigallocatechin gallate.

**Figure 5 ijms-23-09759-f005:**
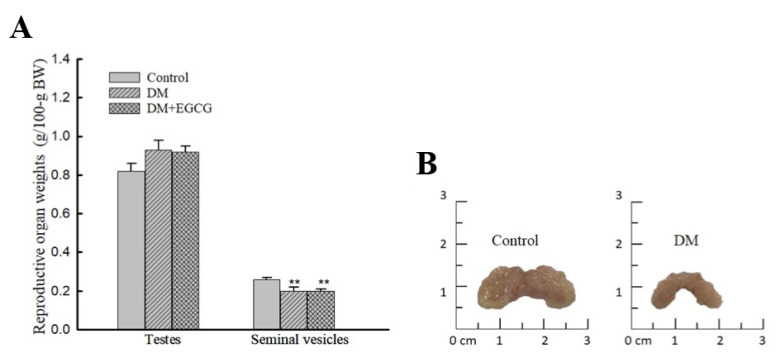
Weights of testes and seminal vesicles (**A**) and photographs of the seminal vesicles of representative control and diabetic rats (**B**). The organ weights are expressed as g/100 g body weight. ** *p* < 0.01 vs. control group. DM, diabetes mellitus; *EGCG*, epigallocatechin gallate.

**Figure 6 ijms-23-09759-f006:**
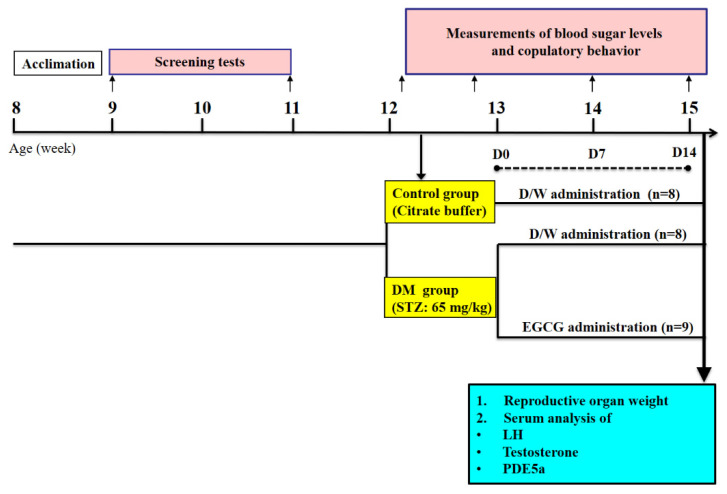
Schematic of study design. After STZ-induced diabetes, animals were treated with 40 mg/kg of EGCG or distilled water. Blood glucose levels and sexual behavior tests were performed before, and after 7 and 14 days of treatment. After the last behavioral test, the animals were sacrificed, and blood was collected for serum LH, testosterone, and PDE5a analysis.

**Table 1 ijms-23-09759-t001:** Effect of EGCG on serum LH, testosterone, and PDE5a levels.

Group	LH (ng/mL)	Testosterone (ng/mL)	PDE5a (ng/mL)
Control	19.49 ± 3.79	18.53 ± 1.69	0.33 ± 0.03
DM	5.94 ± 1.70 *^,^^#^	20.39 ± 1.89	0.80 ± 0.17 **^,^^#^
DM +EGCG	18.50 ± 5.84	16.07 ± 2.09	0.37 ± 0.02

All data were given as mean ± SEM. * *p* <0.05, ** *p* < 0.01 vs. control group; ^#^
*p* < 0.05 vs. DM + EGCG group. DM, diabetes mellitus; *EGCG*, epigallocatechin gallate; LH, luteinizing hormone; PDE5a, phosphodiesterase-5a.

## Data Availability

The data that support the findings of this study are available from the corresponding author upon reasonable request.

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
