# Peer review of "Protective Effects of Epigallocatechin Gallate for Male Sexual Dysfunction in Streptozotocin-Induced Diabetic Rats"

_ijms, 2022, doi:10.3390/ijms23179759_

Round 1

Reviewer 1 Report

The Huang et al., 2022, Manuscript ID: ijms- 1861466 addresses the protective role of Epigallocatechin Gallate on male sexual dysfunction in STZ-Induced diabetic rats. A search on Pubmed.gov for the terms "Epigallocatechin Gallate" and "diabetes” and “Sexual” keywords resulted in no hits that depicts the novelty of this study. There are few studies which shows the role of epigallocatechin gallate and male reproductive functions which shows mostly the antioxidative effects of epigallocatechin in STZ-treated testis. There are few important queries and few suggestion which makes this manuscript more representable to be publish.

1.       What are the criteria for the selection of STZ-treated rat as not all the rats develop diabetes? The values of glucose in the serum need to be mentioned in the methods section. You can cite the literature “10.1016/j.biochi.2019.10.014”.

2.       Can the authors show the histomorphological changes in the testis of the mice? The degradative changes in the male sexual organs need to be checked. Follow and cite the article “doi:10.1016/j.bbadis.2018.11.019”.

  1. The poor sexual activity as well as STZ-induced diabetes both has negative impact on T level in the mice [https://doi.org/10.1177/1933719118770547]. In the current study it is not the case, Can the authors justify? The authors may cite the reference.

4.       Do the authors have any results for fasting insulin level in the Epigallocatechin Gallate treated diabetic rat serum? If there, can they cite in the results?

5.       It will be nice to check the molecular mechanism of epigallocatechin gallate effect on diabetic male sexual functions. Do the authors have future study plan to check the molecular signaling? 

Author Response

Point 1.   What are the criteria for the selection of STZ-treated rat as not all the rats develop diabetes? The values of glucose in the serum need to be mentioned in the methods section. You can cite the literature “10.1016/j.biochi.2019.10.014”.

Response : Thank you. We appreciate your concern very much.. We have described diabetes’ rats criteria in section 4.4:

4.4. Induction of diabetes and EGCG treatment

………..Rats exhibiting fasting glucose levels greater than 300 mg/dL after STZ injection were considered as diabetic rats, and were used in this study.” 

Furthermore, wee will also cite the similar study to support our criteria. Thank you.

Point 2.  Can the authors show the histomorphological changes in the testis of the mice? The degradative changes in the male sexual organs need to be checked. Follow and cite the article “doi:10.1016/j.bbadis.2018.11.019”.

Response : Thank you very much for your remark. We have gross image of the seminal vesicles and was shown in figure 5. Markable changes in seminal vesicle size were seen. However, we have no histomorphological images to show. We will cite the article you provided. Thank you for your help.

Point 3.  The poor sexual activity as well as STZ-induced diabetes both has negative impact on T level in the mice [https://doi.org/10.1177/1933719118770547]. In the current study it is not the case, Can the authors justify? The authors may cite the reference.

Response: Thank you for your valuable suggestion. Through the literature: [https://doi.org/10.1177/1933719118770547], we understood obesity and adipokines also have impacts on reproductive system as well as T level. There are some relationships with our article; we will cite this article for readers in our discussion:

……. Body weights were dramatically reduced in both groups of STZ-treated rats, with or without EGCG. However, it is interesting to note that diabetic rats treated with EGCG for 14 days had significantly lower mean body weight than non-treated diabetic ones. Our result supports a previous study showing that EGCG has an anti-obesity effect…………..

Point 4.  Do the authors have any results for fasting insulin level in the Epigallocatechin Gallate treated diabetic rat serum? If there, can they cite in the results?

 Response: We are grateful for your suggestion. However, we did not measure fasting insulin levels in the present study. As your suggestion, we will do it in our further study. 

Point 5.   It will be nice to check the molecular mechanism of epigallocatechin gallate effect on diabetic male sexual functions. Do the authors have future study plan to check the molecular signaling? 

Response: We agree with your comment and suggestion, thank you. Actually, we have the same idea to perform this experiment in our further study.

Reviewer 2 Report

Study is interesting however manuscript is carelessly edited. First of all, formatting of the text should be in accordance with Molecules journal. Why different fonts (italic/bold) are used in manuscript? Moreover, the manuscript should be corrected by native speaker because there are some errors in the text (some examples are given below).

The other comments:

1)      Figure 1: it should be moved to Material and Method section. Abbreviation D/W should be explain in Figure legend. Add the number of objects in each group

2)      Line 100: „were not significant difference” – should be: „were not significantly different”

3)      Line 116-117: the abbreviations: ML, IL, FL should be explained before the first use in the text.

4)      Figure 5 is not cited in the text. The quality of the Figure should be improved. Fonts are hardly visible. „Liner” – should be „linear”.

5) Line 256: reedit the sentence: „STZ (Sigma) was dissolved (…) at a dose of 65 mg/kg body weight.”

Author Response

Point 1. Figure 1: it should be moved to Material and Method section. Abbreviation D/W should be explain in Figure legend. Add the number of objects in each group

Response: Thank you for the suggestions. We have done in the revised manuscript.  The number of the figures have been re-arranged.

Point 2. Line 100: „were not significant difference” – should be: „were not significantly different”

Response: Thank you for your correction.  The above-mentioned points have been made.

Point 3. Line 116-117: the abbreviations: ML, IL, FL should be explained before the first use in the text.

Response: Thank you for your comments.  The above-mentioned points have been made.

Point 4. Figure 5 is not cited in the text. The quality of the Figure should be improved. Fonts are hardly visible. „Liner” – should be „linear”.

Response: Thank you for your suggestions and correction.  The above-mentioned points have been made.

Point 5. Line 256: reedit the sentence: „STZ (Sigma) was dissolved (…) at a dose of 65 mg/kg body weight.”

Response: Thank you for your concern.  Based on this comment, we have modified the following sentence in the revised manuscript:

Section: Materials and methods (P. 9, line no. 4-5)

“STZ (Sigma) was freshly dissolved in 0.1 M sodium citrate buffer (pH 4.5) at a dose of 65 mg/kg body weight”.

Round 2

Reviewer 1 Report

The authors has justified all the comments. 

Author Response

few suggestion which makes this manuscript more representable to be publish.

Point 1.   What are the criteria for the selection of STZ-treated rat as not all the rats develop diabetes? The values of glucose in the serum need to be mentioned in the methods section. You can cite the literature “10.1016/j.biochi.2019.10.014”.

Response 1: Thank you. We appreciate your concern very much.. We have described diabetes’ rats criteria in section 4.4:

4.4. Induction of diabetes and EGCG treatment

………..Rats exhibiting fasting glucose levels greater than 300 mg/dL after STZ injection were considered as diabetic rats, and were used in this study.” 

Furthermore, we will also cite the similar study to support our criteria. Thank you.

Point 2.  Can the authors show the histomorphological changes in the testis of the mice? The degradative changes in the male sexual organs need to be checked. Follow and cite the article “doi:10.1016/j.bbadis.2018.11.019”.

Response 2: Thank you very much for your remark. We have gross image of the seminal vesicles and was shown in figure 5. Markable changes in seminal vesicle size were seen. However, we have no histomorphological images to show. We will cite the article you provided. Thank you for your help.

Point 3.  The poor sexual activity as well as STZ-induced diabetes both has negative impact on T level in the mice [https://doi.org/10.1177/1933719118770547]. In the current study it is not the case, Can the authors justify? The authors may cite the reference.

Response 3: Thank you for your valuable suggestion. Through the literature: [https://doi.org/10.1177/1933719118770547], we understood obesity and adipokines also have impacts on reproductive system as well as T level. There are some relationships with our article; we will cite this article for readers in our discussion:

……. Body weights were dramatically reduced in both groups of STZ-treated rats, with or without EGCG. However, it is interesting to note that diabetic rats treated with EGCG for 14 days had significantly lower mean body weight than non-treated diabetic ones. Our result supports a previous study showing that EGCG has an anti-obesity effect…………..

Point 4.  Do the authors have any results for fasting insulin level in the Epigallocatechin Gallate treated diabetic rat serum? If there, can they cite in the results?

 Response 4: We are grateful for your suggestion. However, we did not measure fasting insulin levels in the present study. As your suggestion, we will do it in our further study. 

Point 5.   It will be nice to check the molecular mechanism of epigallocatechin gallate effect on diabetic male sexual functions. Do the authors have future study plan to check the molecular signaling? 

Response 5: We agree with your comment and suggestion, thank you. Actually, we have the same idea to perform this experiment in our further study.
